# Runtime-Adaptive Pruning for LLM Inference

## Abstract

Large language models (LLMs) excel at language understanding and generation, but their enormous computational and memory requirements hinder deployment. Compression offers a potential solution to mitigate these constraints. However, most existing methods rely on fixed heuristics and thus fail to adapt to runtime memory variations or heterogeneous KV cache demands arising from diverse user requests. To address these limitations, we propose *RAP*, an elastic pruning framework driven by reinforcement learning (RL) that dynamically adjusts compression strategies in a runtime-aware manner. Specifically, *RAP* dynamically tracks the evolving ratio between model parameters and KV-cache across practical execution. Recognizing that FFNs house most parameters, whereas parameter-light attention layers dominate KV-cache formation, the RL agent retains only those components that maximize utility within the current memory budget, conditioned on instantaneous workload and device state. Extensive experiments results demonstrate that *RAP* outperforms state-of-the-art baselines, marking the first time to jointly consider model weights and KV cache on the fly. Anonymous source code is submitted with the paper and will be publicly available.

## 1 Introduction

Large language models (LLMs) has revolutionized artificial intelligence through unprecedented performance in complex language tasks (Brown et al., 2020; Achiam et al., 2023; microsoft; github). The autoregressive architectures, however, pair "billion-parameter" with memory-intensive key–value (KV) caches, inflating both computation and memory footprints (Fedus et al., 2022; Patterson et al., 2021; Touvron et al., 2023b; Chowdhery et al., 2023; Team et al., 2024). While cloud solutions mitigate some burdens, emerging edge scenarios, mobile devices and real-time services (Yuan et al., 2023; Lin et al., 2022; 2024), demand on-device inference that current LLMs cannot sustain. Model compression is widely used to preserve generative quality while slashing resource costs.

To address LLM deployment bottlenecks, three main compression families have emerged: model pruning (Ma et al., 2023b; Zhong et al., 2024; Sun et al., 2024; Shao et al., 2024), knowledge distillation (Sun et al., 2019; Xu et al., 2024; Chen et al., 2024), and quantization (Liu et al., 2024; Lin et al., 2024). We focus on pruning. Existing schemes(Ma et al., 2023b; Zhong et al., 2024; Sun et al., 2024; Shao et al., 2024; Ashkboos et al., 2024; Gao et al., 2024; Men et al., 2024; He et al., 2024; Jaiswal et al., 2024), whether element-, block-, or layer-wise, achieve impressive parameter reductions but assume static workloads and rely on heuristic policies, neglecting runtime variability, as shown in Figure 1. Such rigidity overlooks two dominant sources of autoregressive inference runtime variance: 1) Input-driven variance: batch size and sequence length directly scale the KV cache memory (e.g., Llama-7B (Touvron et al., 2023a) requires 32 GB of KV cache memory, batch = 16 and length = 4k tokens, dwarfing the static 14 GB model parameters. 2) System-level variance. Edge devices often exhibit stochastic runtime variance, for instance, interference from co-running applications, affecting available memory budgets on the fly. This situation presents a compelling research question:

*How to select optimal LLM pruning policy that can adapt to heterogeneous, time-varying request workloads while satisfying fluctuating memory budgets?*

In this paper, we propose *RAP*, a runtime-adaptive pruning framework that addresses these challenges. *RAP* abandons static one-size-fits-all compression in favor of dynamically adjusting the model's sparsity level for each inference. As shown in Figure 1, it introduces a reinforcement learning (RL) agent that observes real-time signals, such as input sequence length, batch characteristics, and current

Figure 1: Illustration of *RAP*. (a) Conventional pruning relies on hand-developed heuristics that focus solely on model weights. (b) *RAP* employs a runtime-adaptive RL agent that dynamically prunes LLMs based on real-time user requests and memory budget constraints.

memory availability, and selects an appropriate pruning policy on the fly. This design ensures that the model stays within memory budgets under tight conditions while preserving as many parameters as possible when resources allow. By coupling compression decisions with the execution context, *RAP* effectively accommodates heterogeneous workloads and fluctuating system constraints that are impractical for fixed pruning strategies. We formulate adaptive pruning as a sequential decision process and train the RL agent to maximize efficiency without compromising output quality. The agent's reward function balances memory savings against generation fidelity, encouraging policies that reduce memory usage only to the extent they do not degrade performance. Once trained, the agent serves as an intelligent controller during inference, guiding the LLM to prune different components (e.g., attention heads, feed-forward channels, or even entire layers) in response to each request's needs. Notably, *RAP* adds negligible runtime cost, since the learned policy can rapidly compute pruning decisions. This yields a flexible, context-aware compression mechanism that seamlessly scales LLM deployments to edge environments. Our experiments demonstrate that *RAP* outperforms static pruning baselines across a range of deployment scenarios. Without manual retuning, *RAP* adapts to varying batch sizes and sequence lengths, consistently meeting fluctuating memory limits while maintaining strong task performance. For example, under stringent memory constraints, *RAP* prunes a substantial fraction of the model's weights to fit an LLM on-device yet maintains accuracy comparable to an unpruned model. Conversely, when memory is abundant, *RAP* leaves the model largely intact to maximize accuracy, effectively achieving the best of both worlds. In summary, our contributions are as follows:

• We propose *RAP*, a novel runtime-adaptive LLM pruning framework that dynamically adjusts model size based on real-time input demands and memory constraints.

• We cast the pruning policy selection as a reinforcement learning problem and develop an RL agent that learns an optimal policy balancing memory efficiency and model fidelity.

• We demonstrate through extensive experiments that *RAP* consistently outperforms static compression strategies under dynamic workloads, achieving superior memory savings and faster inference with minimal impact on output quality.

## 2 BACKGROUND AND RELATED WORK

### 2.1 RUNTIME LLM INFERENCE MEMORY BREAKDOWN

Transformer-based LLMs comprise a stack of *homogeneous* decoder layer, each with a multi-head attention (MHA) block followed by a feed-forward network (FFN) block. Given that FFNs typically contain approximately $2\times$ the parameters of their corresponding attention modules, the static parameter memory allocation is predominantly determined by FFN weights, which remain fixed once model are loaded. During inference, each token $x$ is projected with $W_q$, $W_k$, and $W_v$ within MHA to obtain $Q = xW_q$, $K = xW_k$, and $V = xW_v$; the resulting $K$ and $V$ tensors are appended to the KV cache across all layers. For Llama2-7B ($n_{\text{layers}} = 32$, $n_{\text{heads}} = 32$, $d_{\text{head}} = 128$), the per-token cache cost is $\text{Memory}_{\text{KV}} = 2\, n_{\text{layers}}\, n_{\text{heads}}\, d_{\text{head}}\, p_a \approx 0.5$ MB, where the factor 2 stores both keys and values. Figure 3 shows memory footprint across batch size and sequence length. Each pie chart illustrates the relative proportion of memory consumed by model parameters (FFN in orange, MHA in blue) and KV cache (gray). As batch size and sequence length gradually extend, memory consumption transitions from parameter-dominated regimes to KV cache-dominated, highlighting the dynamic

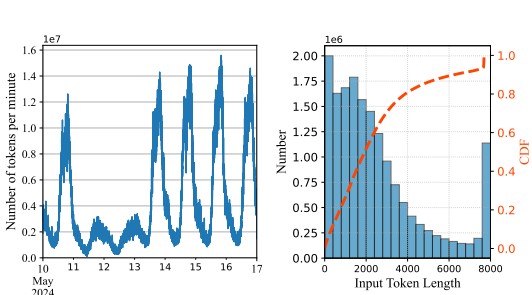

Figure 2: Distribution and daily variation of a conversational LLM inference workload.

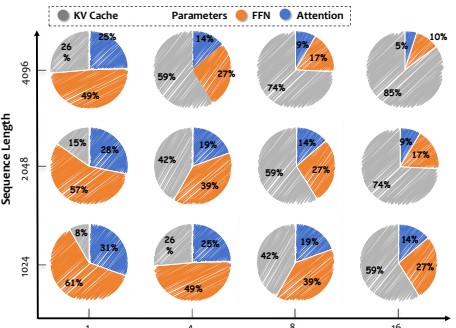

Figure 3: Dynamic memory footprint across varying batch sizes and sequence lengths.

nature of memory bottlenecks in practical deployment. Once model is loaded into memory, increasing the batch size or extending the context length does not affect parameter memory consumption but substantially increases KV cache memory overhead.

$$\text{KV cache} \ \propto \ (\text{batch size}) \times (\text{sequence length}) \times n_{\text{layers}}. \tag{1}$$

Therefore, practical memory scaling is driven almost entirely by the MHA-generated KV cache, underscoring the need for adaptive compression schemes that address both the FFN-heavy static parameter and this rapidly expanding dynamic cache.

## 2.2 EXISTING LLM PRUNING

For runtime LLM inference, pruning strategies (Hu et al., 2021; Liu et al., 2023; Xia et al., 2023; Yin et al., 2023; Zhang et al., 2023) must balance efficiency, accuracy, and adaptability. **1) Static vs. dynamic pruning:** Static methods (e.g., ISC (Das et al., 2023), SparseGPT (Frantar & Alistarh, 2023), E-Sparse (Li et al., 2023), Wanda (Sun et al., 2024)) apply fixed sparsity without retraining, achieving up to 50% sparsity but degrading under higher sparsity levels and fundamentally lacking adaptability. Structured variants (Ashkboos et al., 2024; Chen et al., 2023; Ma et al., 2023b; Zhao et al., 2024) improve hardware efficiency but require retraining (e.g., LoRA (Hu et al., 2021)). In contrast, dynamic pruning (An et al., 2024; Federici et al., 2024; Le et al., 2025; Liu et al., 2023) adapts per input, improving flexibility but retaining full weights and inducing irregular sparsity, limiting hardware speedups. **2) Parameter-only vs. parameter+KV compression:** Most pruning reduces weights (Ma et al., 2023b; Ashkboos et al., 2024; Li et al., 2023; Sun et al., 2024) but ignores KV cache, a major runtime bottleneck. While weight pruning shrinks static parameter, it fails under long-context due to exponentially growing KV cache. Recent methods (e.g., ShortGPT (Men et al., 2024), BlockPruner (Zhong et al., 2024), LLM-Drop (He et al., 2024), FinerCut (Zhang et al., 2024b)) prune both parameters and KV cache, reducing computation and memory but often rely on static rules, sacrificing accuracy. The core trade-off persists: parameter-only pruning is insufficient, while aggressive KV cache pruning hurts performance. **3) Heuristic vs. learning-based control:** Heuristic methods (Sun et al., 2024; Frantar & Alistarh, 2023; Ma et al., 2023b) use static scores (e.g., magnitude, saliency), lacking runtime adaptability or end-to-end optimization. Learning-based policies can jointly optimize for speed, memory, and accuracy. Though RL has proven effective in (Andrychowicz et al., 2020; Mnih et al., 2015; Zhang et al., 2017), it remains underexplored for LLM pruning, particularly for coordinated control of parameter and KV cache. *RAP* addresses this by introducing an RL-based policy that dynamically prunes both components, enabling runtime-adaptive and efficient inference beyond static baselines.

## 3 MOTIVATION

In this section, we present key observations from stochastic workloads, model-intrinsic and system factors for practical LLM inference.

▶ Takeaway 1: *Runtime workloads are inherently dynamic.* Modern LLM service platforms must operate under highly volatile workload conditions (Patel et al., 2024; Li et al., 2024; Jaiswal et al., 2025). Figure 2, derived from an Azure LLM-inference trace (Stojkovic et al., 2025), reveals that prompt-length distributions and request arrival rates fluctuate markedly over time, producing a

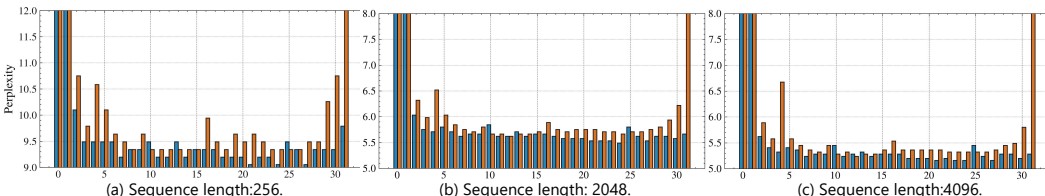

Figure 4: Block sensitivity analysis: removing specific MHA and FFN under diff. sequence length.

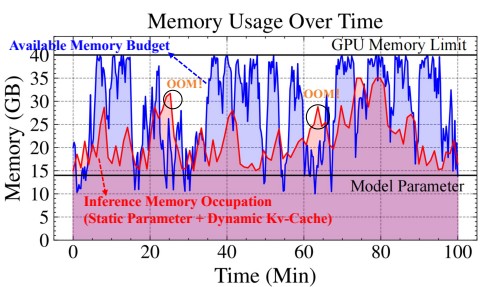

Figure 5: Dynamic memory allocation trace for Llama2-7B on an NVIDIA A40 (40 GB) (NVIDIA Corporation, 2020). Blue indicates available memory; red shows real-time usage (model + KV cache), which scales with workload and cause out-of-memory (OOM) errors under heavy requests.

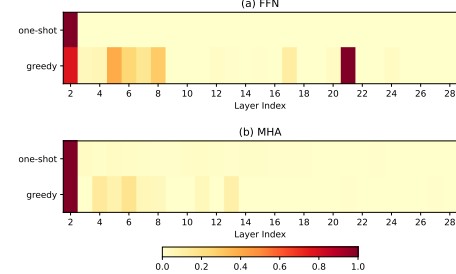

Figure 6: Per-block perplexity sensitivity (FFN vs. MHA) under one-shot and GSI pruning, with GSI revealing inter-layer heterogeneity missed by static one-shot methods.

non-deterministic mix of short conversational turns and burst, long-form inputs. Figure 3 illustrates how memory allocation transitions from parameter-dominated regimes at low batch sizes to KV cache-dominated scenarios as batch size and sequence length scale up, fundamentally reshaping inference memory bottlenecks. These findings reveal a fundamental limitation in current serving infrastructures: static resource allocation and heuristic per-request throttling mechanisms fail to satisfy Quality of Experience (QoE) demands for latency and memory efficiency when facing the inherently dynamic and unpredictable nature of real-time inference workloads.

▶ Takeaway 2: *Homogeneous blocks exhibit heterogeneous impact.* Current transformer architectures exhibit seemingly homogeneous layers (§2.1), yet their internal blocks (MHA and FFN) contribute heterogeneously to generation quality. Prior studies have broadly differentiated layer importance (Ma et al., 2023a; Zhang et al., 2024a; Yao et al., 2024; Pan et al., 2024) or assumed fixed superiority of FFN over MHA (He et al., 2024). However, as summarized in Figure 4, the impact of MHA and FFN removal on perplexity (PPL) varies significantly across layers, challenging coarse-grained or uniform assumptions. Moreover, solely optimizing FFN cannot alleviate the KV cache bottleneck that arises in long sequences and large batch sizes. Additionally, block importance demonstrates dynamic shifts across different request lengths, underscoring the limitations of existing static, heuristic-based pruning strategies (Yao et al., 2024; Pan et al., 2024; Ma et al., 2023a; Men et al., 2024). These insights highlight the critical imperative for adaptive method that dynamically discerns and leverages block-level sensitivity to accommodate heterogeneous runtime computational demands.

▶ Takeaway 3: *Real-world systems demonstrates runtime variance.* Real-world LLM inference systems rarely maintain consistent memory availability (Wang et al., 2024; Yu et al., 2023; Xu et al., 2022). Instead, they encounter dynamic memory variability driven by heterogeneous LLM workloads and interference by co-running applications. Azure LLM service traces (Stojkovic et al., 2025) show that prompt-length and request-arrival spikes can induce instantaneous GPU-memory fluctuations of up to 5–10. Concurrent workloads further disrupt cache and bandwidth allocation, exacerbating contention and latency instability. As Figure 5 illustrates, these memory surges often preempt co-running applications and invalidate the fixed-budget assumptions of existing pruning and scheduling methods, highlighting a critical gap between current serving frameworks and real-world, memory-dynamic inference environments.

## 4    *RAP* DESIGN

In this section, we present the design of *RAP*. Specifically, we first introduce greedy sequential importance analysis §4.1 to thoroughly assess the impact of individual transformer blocks. Then, we explain how we formulate the problem of runtime dynamic pruning as an RL task §4.2.

## 4.1 Greedy Sequential Importance

As discussed in §3, blocks exhibit heterogeneous contributions to model performance. Conventional one-shot pruning methods (Ma et al., 2023b; Zhong et al., 2024) that remove layers solely based on individual sensitivity ignore inter-layer dependencies, often leading to substantial performance degrades under aggressive compression ratio. The deep composition of nonlinear activations and residual connections in LLMs induces strong inter-layer dependencies (Ling et al., 2024; Meng et al., 2024), rendering the network fragile to architectural changes. Consequently, excising even a single layer can trigger a cascade of representational errors that corrupts the entire model's functionality. To mitigate this, we propose *Greedy Sequential Importance* (GSI) analysis. As detailed in Algorithm 1, GSI performs iterative pruning by progressively removing the block whose exclusion results in the minimal deterioration, followed by re-evaluating after each step. This step-wise recalibration controls error accumulation, stabilizes accuracy over successive pruning stages, and achieves a more balanced compression–performance trade-off compared to static, one-shot methods. Figure 6 further highlights that one-shot pruning neglects inter-layer heterogeneity, leading to suboptimal pruning decisions. In this paper, we select perplexity as the proxy metric for the GSI algorithm to measure the impact of block removal on overall model performance, since perplexity is a widely-accepted metric for generative capabilities of LLM. Alternatively, task-specific downstream metrics can serve as a proxy to enable pruning decisions more aligned with target scenarios. Overall, GSI offers a principled and adaptive approach to LLM compression, effectively balancing model size reduction with task performance preservation.

---

**Algorithm 1** Greedy Sequential Importance (GSI) using perplexity as the proxy metric

---

**Require:** Pre-trained model $\mathcal{M}$, evaluation corpus $\mathcal{C}$, target prune ratio $\rho$
**Ensure:** Pruned model $\mathcal{M}^{(t)}$, pruned blocks $\{B_{k_t}\}$, perplexities $\{P_{k_t}\}$

1: $\mathcal{M}^{(0)} \leftarrow \mathcal{M}, \quad t \leftarrow 0$ ▷ Initialization
2: **while** $\mathrm{PruneRatio}(\mathcal{M}^{(t)}) < \rho$ **do**
3:      $t \leftarrow t + 1$
4:      **for all** block $B_i$ in $\mathcal{M}^{(t-1)}$ **do**
5:          $\hat{\mathcal{M}}_i \leftarrow \mathcal{M}^{(t-1)} \setminus B_i$ ▷ Candidate Model
6:          $P_i \leftarrow \exp\left(-\frac{1}{|\mathcal{C}|} \sum_{w \in \mathcal{C}} \log p_{\hat{\mathcal{M}}_i}(w)\right)$ ▷ Perplexity
7:      **end for**
8:      $k \leftarrow \arg\min_i P_i$ ▷ Greedy Selection
9:      $\mathcal{M}^{(t)} \leftarrow \hat{\mathcal{M}}_k$ ▷ Model Update
10: **end while**
11: **return** $\mathcal{M}^{(t)}, \{B_{k_1}, \ldots, B_{k_t}\}, \{P_{k_1}, \ldots, P_{k_t}\}$

---

## 4.2 RL-Guided Pruning Decisions

To address the dynamic inference environments characterized by user-request workloads and system runtime variance, we propose *RAP* an adaptive pruning framework based on reinforcement learning. Figure 7 presents the design overview of *RAP*. ① At each inference step, *RAP* observes the real-time request characteristic, model configuration, and available memory budget to determine the current execution state. ② Based on this state, the RL agent selects a pruning policy that satisfies the memory constraint while aiming to preserve model performance. ③ The base model then executes the selected pruning policy by removing the corresponding FFN and MHA blocks, and subsequently performs inference on the compressed architecture. ④ Finally, *RAP* evaluates inference metrics, including memory overhead and perplexity, to derive a reward that quantifies how effectively the selected action balances computational efficiency with model performance. We next define the core RL components, *State*, *Action*, and *Reward*, to formalize the optimization space of *RAP*.

**State:** the state at the $t$-th timestep $s_t = (s_t^{Req}, s_t^{Model}, s_t^{Sys}) \in \mathbf{S}$ consists of three components:

- $s_t^{Req} = (R_{bs}, R_{sql})$ captures the real-time request characteristics, consisting of the batch size $R_{bs}$ and sequence length $R_{sql}$.
- $s_t^{Model} = (\{\mathrm{MHA}_{imp,i}\}_{i=1}^N, \{\mathrm{FFN}_{imp,i}\}_{i=1}^N)$, representing importance score of each MHA and FFN block computed via GSI algorithm §4.1, where $N$ denotes the total number of blocks, capturing the granular block-level model configuration.

Figure 7: Design overview of *RAP*. (1) Runtime statistics from inference environment are encoded into execution state. (2) RL agent selects FFN/MHA blocks for pruning. (3) Resulting memory consumption and performance constitute the reward. (4) Agent gains reward and reinforces, completing an online loop for dynamically balanced efficiency and accuracy.

- $s_t^{Sys} = (Sys_{avail}, Sys_{req})$ represents the runtime system memory state, where $Sys_{avail}$ denotes the currently available system memory, and $Sys_{req}$ indicates the anticipated memory overhead after executing the selected pruning policy.

**ACTION:** At each pruning step $t$, given $N$ layers model and input state $s_t$, the action set is defined as $A_t = \{(a_1, \ldots, a_{2N}) \mid a_i \in \{0, 1\}\}$, where $2N$ binary indicators represent whether to retain ($a_i = 1$) or remove ($a_i = 0$) each of the $2N$ transformer blocks (one MHA and one FFN per layer) at step $t$. Simultaneous multi-block selection creates an intractable action space of size $2^{2N}$; for example, Llama2-7B with 64 blocks, this yields approximately $2^{64} \approx 1.8 \times 10^{19}$ possible actions. To address this computational challenge, we decompose the decision into sequential single-block selections, reducing the action space to $2N$ decision step. However, directly applying one-shot top-$k$ pruning proves suboptimal, as demonstrated in § 4.1, since block importance dynamically changes after each removal. More precisely, we utilize GSI-derived importance scores to iteratively remove the least important block at each step. After each removal, we re-assess the importance hierarchy among remaining blocks and select the next least important candidate, repeating this greedy selection until the peak memory footprint meets the memory budget constraint. While the approach requires iterative decision-making, the RL agent employs a lightweight 2-layer MLP with minimal parameters, ensuring computational overhead remains negligible compared to the inference costs of billion-parameter LLMs.

**REWARD:** To ptimize a pruned model under a fixed memory budget presents a multi-objective challenge. We address this by formulating a unified reward as a weighted sum of two specialized metrics: $R_{\text{ppl}}$, reflecting language modeling performance via perplexity, and $R_{\text{mem}}$, which penalizes peak memory consumption during inference.

$$R_t = \sum_{i=1}^{2N} (A_t)_i \left( \alpha R_i^{\text{ppl}} - \beta R_i^{\text{mem}} \right) \quad (2)$$

Here, $N$ is the total number of blocks. At each time step $t$, $A_t$ is a binary action vector where $(A_t)_i = 1$ indicates that block $i$ is preserved while $(A_t)_i = 0$ will remove block $i$. The terms $R_i^{\text{ppl}}$ (detailed in § 4.2) and $R_i^{\text{me}}$ denote the perplexity importance and estimated memory footprint of block $i$, respectively. The hyperparameters $\alpha$ and $\beta$ act as reward scale factors, tuning the accuracy–memory trade-off and penalizing bottleneck workloads when necessary. Specifically, in this paper, we set $\alpha = 1$, $\beta = 0.3$. A detailed description of RL-agent algorithm can be found in Appendix A.

## 5 EXPERIMENTS

### 5.1 EXPERIMENTAL SETUP

**Model and Dataset.** We implemented *RAP* using PyTorch (Paszke et al., 2019) and the HuggingFace Transformers library (Wolf et al., 2019) for managing models and datasets. All experiments were conducted on NVIDIA A40 GPUs (NVIDIA Corporation, 2020). For GSI, we used the Alpaca

Table 1: Zero-shot performance of pruned versus dense model under different memory budgets. [1] 100% memory budget indicates exceeding peak inference memory usage (parameters + KV cache).

| Budget | Schemes | Perplexity ↓ | | Commonsense Task (%) ↑ | | | | | | | |
| | | WikiText2 | PTB | BoolQ | PIQA | WinoG. | HellaS. | ARC-e | ARC-c | OBQA | Avg. |
|---|---|---|---|---|---|---|---|---|---|---|---|
| | | | | **Llama2-7B** | | | | | | | |
| 100%[1] | Dense | 5.47 | 24.09 | 77.74 | 79.11 | 68.97 | 75.98 | 74.62 | 46.25 | 44.20 | 66.70 |
| 80% | LLMPruner (Ma et al., 2023b) | 28.42 | 278.05 | 50.63 | 68.82 | 54.14 | 52.66 | 49.33 | 30.20 | 34.59 | 48.63 |
| | SliceGPT (Ashkboos et al., 2024) | 58.33 | 211.33 | 61.99 | 65.29 | 58.08 | 43.43 | 52.27 | 32.08 | 28.99 | 48.88 |
| | ShortGPT (Men et al., 2024) | 79.49 | 171.02 | 62.17 | 60.12 | 60.38 | 43.70 | 41.25 | 30.12 | 35.00 | 47.53 |
| | MHA-Drop (He et al., 2024) | 1068.99 | 1579.65 | 46.08 | 54.56 | 50.83 | 29.12 | 28.62 | 27.47 | 25.20 | 37.41 |
| | FFN-Skip (Jaiswal et al., 2024) | 28720.72 | 32216.40 | 42.80 | 49.35 | 51.22 | 26.69 | 27.18 | 28.49 | 26.60 | 36.05 |
| | *RAP* | **11.80** | **46.56** | **62.81** | **73.99** | **63.38** | **65.77** | **60.35** | **36.69** | **36.60** | **57.08** |
| 60% | LLMPruner | 96.52 | 711.38 | 55.96 | 61.15 | 50.83 | **38.09** | 35.06 | 27.13 | 28.40 | 42.38 |
| | SliceGPT | 348.26 | 590.12 | **61.04** | 54.56 | 49.49 | 28.99 | 30.68 | 23.46 | 25.40 | 39.09 |
| | ShortGPT | 964.92 | 2219.93 | 55.57 | 50.98 | 50.51 | 27.83 | 26.39 | 27.47 | 26.60 | 37.91 |
| | MHA-Drop | 6731.72 | 7914.86 | 37.83 | 49.89 | 49.88 | 25.72 | 26.05 | 25.43 | 26.80 | 34.51 |
| | FFN-Skip | 202008.00 | 160986.48 | 44.83 | 50.64 | 48.60 | 25.86 | 25.72 | 28.92 | 28.19 | 36.13 |
| | *RAP* | 84.78 | 592.65 | 57.16 | 58.26 | 53.75 | 37.81 | **37.79** | 26.02 | **30.20** | **43.00** |
| | | | | **Llama3-8B** | | | | | | | |
| 100% | Dense | 6.13 | 9.91 | 81.35 | 80.78 | 72.61 | 79.14 | 77.69 | 53.33 | 45.00 | 69.99 |
| 80% | LLMPruner | 48.94 | 99.33 | 62.17 | 65.02 | 51.93 | 42.32 | 41.33 | 25.50 | 29.40 | 45.37 |
| | SliceGPT | 143.93 | 71.99 | 61.87 | 65.94 | 54.05 | 42.51 | 54.37 | 31.06 | 27.80 | 48.23 |
| | ShortGPT | 37412.61 | 41988.11 | 56.82 | 58.59 | 54.85 | 37.71 | 36.99 | 30.20 | 28.00 | 43.30 |
| | MHA-Drop | 529.00 | 737.98 | 37.80 | 53.92 | 50.20 | 26.87 | 30.47 | 23.81 | 27.20 | 35.75 |
| | FFN-Skip | 164387.73 | 149698.12 | 55.77 | 51.47 | 50.74 | 25.90 | 25.75 | 25.25 | 28.00 | 37.56 |
| | *RAP* | **12.98** | **27.15** | **68.84** | **76.55** | **66.11** | **65.55** | **62.12** | **39.51** | **39.60** | **59.77** |
| 60% | LLMPruner | 4009.81 | 1882.99 | 40.48 | 50.05 | **52.33** | 26.26 | 26.77 | 25.94 | 27.00 | 35.55 |
| | SliceGPT | 2844.28 | 1084.03 | 40.86 | 53.86 | 48.93 | 27.97 | 32.07 | 23.04 | 25.80 | 36.08 |
| | ShortGPT | 13284.81 | 13512.55 | 41.44 | 50.98 | 50.03 | 26.74 | 25.46 | 25.34 | 26.80 | 35.26 |
| | MHA-Drop | 1757.11 | 2102.15 | 37.83 | 51.90 | 50.12 | 26.04 | 27.86 | 22.95 | 25.80 | 34.64 |
| | FFN-Skip | 624965.25 | 765475.19 | 51.93 | 51.95 | 50.03 | 26.02 | 24.03 | **26.96** | 28.59 | 37.08 |
| | *RAP* | 246.53 | 355.47 | 52.20 | 56.69 | 50.36 | 31.91 | 33.54 | 24.23 | 27.40 | 39.48 |

dataset (Taori et al., 2023) to compute perplexity importance. We evaluated *RAP* over representative LLMs: Llama2-7B (Touvron et al., 2023c), Llama3-8B (Dubey et al., 2024), Qwen1.5-7B (Bai et al., 2023) and Qwen2.5-7B (Yang et al., 2024). We assessed model performance using the LM Evaluation Harness (Gao et al., 2023) following Llama's evaluation protocol to perform zero-shot task classification on common sense reasoning datasets: BoolQ (Clark et al., 2019), PIQA (Bisk et al., 2020), HellaSwag (Zellers et al., 2019), WinoGrande (Sakaguchi et al., 2019), ARC-easy (Clark et al., 2018), ARC-challenge (Clark et al., 2018), and OpenbookQA (Mihaylov et al., 2018). We tested the model generative ability using WikiText2 (Merity et al., 2016) and PTB (Marcus et al., 1993) dataset. A detailed description of the benchmarks can be found in Appendix B.1.

**Baselines.** To validate the effectiveness of *RAP*, we compared several structured pruning methods: 1) LLMPruner (Ma et al., 2023a): Structural pruning via gradient-weight analysis to remove non-critical coupled layers; omits post-training for fair comparison but incurs pruning-policy overhead. 2) SliceGPT (Ashkboos et al., 2024): PCA-based post-training sparsification reduces embedding dimensions by projecting hidden representations shallow-to-deep. 3) ShortGPT (Men et al., 2024): Layer-pruning reveals redundancy in LLMs by removing redundant layers with minimal performance loss. 4) MHA-Drop (He et al., 2024): Cosine-similarity-guided pruning of entire multi-head attention layers for inference acceleration. 5) FFN-Skip (Jaiswal et al., 2024): Input-adaptive dynamic skipping of FFN layers during decoding for faster generation with negligible quality trade-offs. A detailed description of the baseline models can be found in Appendix B.2. *Notably*, we diverge from previous works by evaluating all methods under identical memory budget, rather than a fixed pruning ratio. We posit that the pruning ratio is a misleading proxy for actual memory footprint, a claim substantiated by our empirical results which reveal a significant discrepancy. This gap arises primarily from the disproportionate memory overhead of runtime KV cache, which parameter counts alone fail to capture. By focusing on a fixed memory budget, our evaluation framework more accurately reflects the constraints of real-world deployment on resource-limited device, yielding more practical and reliable conclusions.

## 5.2 OVERALL PERFORMANCE

In this section, we evaluate LLama2-7B and LLama3-8B under 80% and 60% unified memory budgets, covering both parameters and KV cache. For clarity, 80% memory budget corresponds to 80% of the peak memory footprint of the original, unpruned model, formally expressed as $80\% * \max(\text{param.} + \text{KV cache})$. Sparsity is progressively increased for each method until the total memory overhead meets the target budget constraint. Detail pruning settings for all baselines are

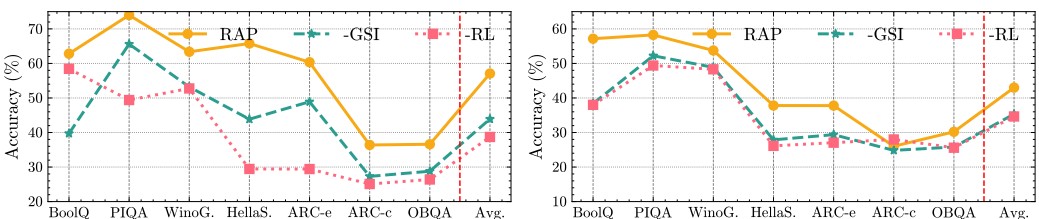

Figure 8: Effectiveness of GSI and RL Agent. Zero-shot performance of $RAP^{-\text{GSI}}$, $RAP^{-\text{RL}}$ versus *RAP* evaluation on Llama2-7B under (a) 80% and (b) 60% memory budgets.

detailed in Table 4 of Appendix C. Evaluated results with the same setting for Qwen1.5-7B and Qwen2.5-7B can be found in Table 3 of Appendix C.

**Generation Ability.** As shown in Table 1, *RAP* shows the smallest perplexity drift among all structured-pruning baselines. Specifically, at the 80% budget perplexity rises by only +6.3 on WikiText2 and +22.6 on PTB, outperforming the next-best method by 16.6 and 9.6, respectively. This advantage persists more pronounced under the harsher 60% cap, stemming from learning an architecturally-aware pruning policy targeting MHA when KV cache is the primary bottleneck and FFN blocks when parameter memory dominates. This asymmetric strategy adapts to the architectural nuances of different models. For instance, the FFN-heavy Llama3-8B (which uses GQA (Ainslie et al., 2023)) is highly sensitive to FFN removal, whereas the standard Llama2-7B is more degraded by pruning MHA. This learned selectivity allows *RAP* to navigate architectural trade-offs, preserving crucial generative capabilities even under severe compression.

**Downstream Task Performance.** We next evaluate zero-shot commonsense reasoning for the same memory budgets. As shown in Table 1, *RAP* again delivers the highest accuracy. Under the 80% budget it preserves 86% of dense accuracy and surpasses the leading baseline by +7.7% on Llama2-7B and +11.5% on Llama3-8B, with the largest gains on HellaSwag and ARC-e. Under an aggressive 60% memory budget, while all methods degrade, *RAP* proves uniquely resilient. It is the sole method to retain over 50% of the dense model's performance, achieving scores of 43.0% on Llama2-7B (0.6% ↑ vs. runner-up) and 39.5% on Llama3-8B (2.4% ↑). These results indicate that *RAP*'s memory-aware, block-level pruning, which considers both parameter and KV cache memory constraints, provides superior performance retention compared to conventional approaches under severe memory limitations.

## 5.3 ABLATION STUDY

To explore the contribution of each component in *RAP*, we design two ablation variants: (1) $RAP^{-\text{GSI}}$, which disables the iterative, Greedy Sequential Importance scorer and instead applies standard static, one-shot perplexity scoring across all requests; and (2) $RAP^{-\text{RL}}$, which removes the RL agent and randomly drops blocks, where '−' means disable or remove proposed module.

**Effectiveness of GSI.** To isolate the contribution of the Greedy Sequential Importance, we implement a static baseline that performs one-shot perplexity evaluation on all blocks initially, then removes the k blocks with the lowest importance to meet the memory budget, without iterative re-evaluation after each removal. Figure 8 and Table 2 reveal that this shortcut severely erodes quality: perplexity on WikiText2 increases to 42.04 and average commonsense accuracy reduces by 13.17%. The degradation arises from latent inter-block dependencies in transformer stacks. Conventional one-shot methods, which score each block independently within the context of the full model, pro-

Table 2: Ablation study on perplexity.

| Budget | Schemes | Perplexity ↓ | |
| | | WikiText2 | PTB |
|---|---|---|---|
| | $\text{MODEL}^{-RL}$ | 313.51 | 535.75 |
| 80% | $\text{MODEL}^{-GSI}$ | 42.04 | 492.97 |
| | **MODEL** | **11.80** | **46.56** |
| | $\text{MODEL}^{-RL}$ | 7249.24 | 9059.14 |
| 60% | $\text{MODEL}^{-GSI}$ | 803.72 | 977.01 |
| | **MODEL** | **74.78** | **592.65** |

duce misleadingly optimistic importance estimates. These estimates become invalid under multi-block pruning scenarios, as they fail to account for inter-block dependencies. GSI addresses this by iteratively pruning the least critical block and then recalibrating the importance of all remaining blocks within the new, contracted architecture. This sequential, state-aware evaluation yields more faithful importance scores, leading to superior performance in high compression regimes.

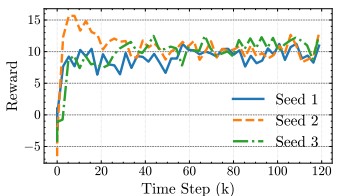

Figure 9: RL reward across different seeds.

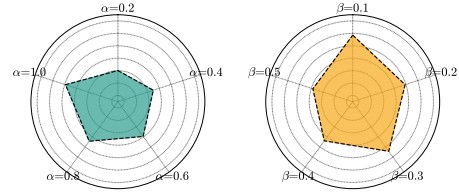

Figure 10: Penalty factors ($\alpha$ and $\beta$) sensitivity.

**Effectiveness of RL Agent.** We next ablate the RL agent that converts GSI scores into real-time actions by comparing *RAP* with a naïve "Random-Drop" baseline that discards the same number of blocks but chooses them uniformly at random. As Figure 8 and Table 2 show, both variants satisfy the memory target, yet *RAP* obviously exceeds the random baseline on generation ability ( -301.71 on WikiText2) and downstream performance ( +18.37%). Crucially, *RAP* accomplishes this online: its RL controller makes block-selection decisions conditioned on the current KV/parameter split, so the policy can tighten or relax MHA/FFN pruning as the request mix shifts. Random-Drop lacks such awareness; each inference call therefore risks violating latency or memory constraints on resource-constrained devices. In short, RL preserves GSI's quality while adding workload-adaptive guarantees, making *RAP* the more practical choice for on-device deployment.

### 5.4 FRAMEWORK ANALYSIS

**Robustness of *RAP*.** To verify that *RAP*'s learning dynamics are not brittle to initialization, we retrain the agent on Llama2-7B with three independent random seeds. Figure 9 plots the seed-wise expected-reward curves. All trajectories increase smoothly and converge within a narrow band, showing that the agent consistently discovers high-quality pruning policies despite stochastic exploration. This stability stems from the architecture introduced above: (i) the Greedy Sequential Importance scorer supplies a well-shaped, low-variance reward signal, and (ii) the memory-aware action mask constrains the search space so early missteps cannot derail policy improvement. Collectively, these components make *RAP*'s reinforcement learning process both robust and generalizable across random seeds.

**Impact of penalty factors $\alpha$ and $\beta$.** *RAP* reward function integrates task utility with two penalty terms, weighted by $\alpha$ and $\beta$, to discourage accuracy degradation (importance decay) and excessive memory usage, respectively. By sweeping $\alpha \in [0.2, 1.0]$ and $\beta \in [0.1, 0.5]$, users can tune the performance–efficiency trade-off to match deployment needs. As shown in Figure 10, higher $\alpha$ values guide the policy to preserve critical blocks, while higher $\beta$ values encourage pruning memory-intensive ones. The optimal reward ridge emerges at large $\alpha$ and moderate $\beta$; we adopt $\alpha=1.0$ and $\beta=0.3$ in all experiments.

**Overhead Analysis.** As shown in Figure 11 in Appendix C. *RAP*'s RL controller adds negligible deployment overhead. While Llama2-7B has $\sim$6.7B parameters and requires 33GB memory for 2048-token inference at batch size 8, the controller has just 18K parameters over $3.7 \times 10^5 \times$ reduction. Latency overhead is negligible: the unpruned model requires 52.73s for inference with sequence length 2048 and batch size 8, whereas a policy step completes in 0.5s ( $< 1\%$ overhead). Even including the one-time 302s offline policy training, the amortized cost is negligible. This efficiency stems from the controller's compact two-layer MLP, which processes Greedy Sequential Importance scores and applies memory-aware masking to accelerate pruning.

## 6 CONCLUSION

This paper addresses the deployment challenges of LLMs caused by their excessive computational and memory demands. While compression techniques have been proposed to mitigate these constraints, existing methods rely on static heuristics and fail to adapt to runtime memory fluctuations or heterogeneous KV cache requirements stemming from diverse user workloads. To overcome these limitations, we introduce *RAP*, an elastic pruning framework powered by RL that dynamically optimizes compression strategies in real-time based on system conditions. This work bridges the gap between static compression techniques and dynamic real-world deployment scenarios, offering a scalable solution for efficient LLM inference in heterogeneous environments.

ETHICS STATEMENT

We affirm adherence to the ICLR Code of Ethics. This work studies compression methods for large language models and does not involve human subjects, personally identifiable information, or sensitive attributes. All datasets and pretrained weights used are publicly available and were accessed and used in accordance with their licenses and terms of use; no data scraping outside the providers' terms was performed. We disclose our use of LLM-based writing assistance in a separate LLM-usage section in Appendix E. Potential risks include lowering the computational barrier for deploying more capable models in resource-constrained settings; to mitigate misuse concerns, we evaluate only on standard public benchmarks, refrain from releasing domain-specific models for sensitive applications, and provide documentation to support responsible use. The authors take full responsibility for the integrity and accuracy of the reported results.

REPRODUCIBILITY STATEMENT

We provide an anonymized artifact in the supplemental materials containing: (i) source code; (ii) configuration files with all hyperparameters; and (iii) step-by-step commands to regenerate all main result. The main paper and appendix details data preprocessing, evaluation metrics, and training/inference procedures, together with hardware specifications and estimated compute budgets. Unless otherwise stated, results are averaged over multiple seeds and we report mean $\pm$ standard deviation; deviations from this protocol are explicitly noted. These materials enable end-to-end reproduction of every quantitative claim in the paper.

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

# A    DETAIL OF RL-AGENT ALGORITHM

## A.1    PROBLEM FORMULATION

We cast RAP as a finite-horizon MDP $\mathcal{M} = (\mathcal{S}, \mathcal{A}, \mathcal{P}, \mathcal{R}, \gamma)$ with horizon $H \leq 2N$, where $N$ is the number of transformer layers and each layer contributes one MHA block and one FFN block (thus $2N$ removable blocks).

**State.**    At decision step $t$, the state $s_t \in \mathcal{S}$ concatenates request-, model-, and system-level features:

$$s_t = \left( s_t^{\text{Req}}, s_t^{\text{Model}}, s_t^{\text{Sys}} \right),$$

with

$$s_t^{\text{Req}} = (R_{\text{bs}}, R_{\text{sql}}), \qquad s_t^{\text{Model}} = \left( \{\text{MHA}_{\text{imp},i}^{(t)}\}_{i=1}^N, \{\text{FFN}_{\text{imp},i}^{(t)}\}_{i=1}^N \right),$$

$$s_t^{\text{Sys}} = (\text{Sys}_{\text{avail}}^{(t)}, \widehat{\text{Sys}_{\text{req}}}^{(t)}).$$

Here $\text{MHA}_{\text{imp},i}^{(t)}$ and $\text{FFN}_{\text{imp},i}^{(t)}$ are the current Greedy Sequential Importance (GSI) scores recomputed after each removal (see Alg. 1); $\text{Sys}_{\text{avail}}^{(t)}$ is the available GPU memory observed at time $t$; and $\widehat{\text{Sys}_{\text{req}}}^{(t)}$ is the agent's estimate of the peak memory after applying the candidate action.

**Action.**    We adopt sequential single-block decisions compatible with DQN:

$$\mathcal{A} = \{0, 1, 2, \ldots, 2N\}.$$

Action $a_t = 0$ denotes STOP; $a_t \in \{1, \ldots, 2N\}$ removes the corresponding block (one of the $N$ MHA or $N$ FFN blocks). An action mask invalidates pruned blocks and can optionally disable actions predicted to break correctness constraints. The episode terminates when either: (i) STOP is taken, or (ii) the peak memory fits the budget.

**Transition.**    Given $(s_t, a_t)$, the environment deterministically updates the pruned architecture $\mathcal{M}_t \mapsto \mathcal{M}_{t+1}$ by excising the selected block if $a_t > 0$, then re-evaluates the GSI scores on the contracted model to produce $s_{t+1}$. Runtime memory availability $\text{Sys}_{\text{avail}}^{(t+1)}$ can be treated as exogenous.

**Discount.**    We set $\gamma = 0.99$.

## A.2    MEMORY MODEL (PEAK GPU FOOTPRINT)

Consistent with the main text, the peak inference memory comprises static parameters and dynamic KV cache. Let $b_{\text{prec}}$ be bytes per scalar (e.g., 2 for bfloat16). For a model state $\mathcal{M}$ (after some blocks are removed) and a request tuple $(R_{\text{bs}}, R_{\text{sql}})$, we estimate

$$\text{Mem}_{\text{param}}(\mathcal{M}) = b_{\text{prec}} \sum_{B \in \mathcal{B}(\mathcal{M})} \#\text{params}(B), \tag{3}$$

$$\text{Mem}_{\text{KV}}(\mathcal{M}, R_{\text{bs}}, R_{\text{sql}}) = b_{\text{prec}} \cdot 2 \sum_{\ell \in \mathcal{L}(\mathcal{M})} n_{\text{heads},\ell} \, d_{\text{head},\ell} \, R_{\text{bs}} \, R_{\text{sql}}, \tag{4}$$

where $\mathcal{B}(\mathcal{M})$ and $\mathcal{L}(\mathcal{M})$ denote remaining blocks and layers, respectively; the factor 2 stores keys and values. The peak is

$$\text{Mem}_{\text{peak}}(\mathcal{M}, R_{\text{bs}}, R_{\text{sql}}) = \text{Mem}_{\text{param}}(\mathcal{M}) + \text{Mem}_{\text{KV}}(\mathcal{M}, R_{\text{bs}}, R_{\text{sql}}).$$

This matches the linear KV-cache scaling with batch and sequence length emphasized in the main paper.

## A.3 DQN-BASED POLICY LEARNING WITH ACTION MASKING

Let $Q_\theta(s, a)$ be the action-value function and $Q_{\bar{\theta}}$ its target copy. We adopt masked $\varepsilon$-greedy:

$$\pi(a|s) = \begin{cases} \text{uniform over valid actions} & \text{with prob. } \varepsilon, \\ \arg\max_{a \in \mathcal{A}_{\text{valid}}(s)} Q_\theta(s, a) & \text{with prob. } 1 - \varepsilon, \end{cases}$$

where $\mathcal{A}_{\text{valid}}(s)$ removes already-pruned blocks and can optionally include feasibility heuristics. With transitions $(s_t, a_t, r_t, s_{t+1}, \text{done})$ stored in replay buffer $\mathcal{D}$, we minimize

$$\mathcal{L}(\theta) = \mathbb{E}_{(s,a,r,s',d) \sim \mathcal{D}} \left[ \left( Q_\theta(s, a) - y \right)^2 \right], \quad y = r + \gamma(1 - d) \max_{a' \in \mathcal{A}_{\text{valid}}(s')} Q_{\bar{\theta}}(s', a').$$

We soft-update the target network periodically: $\bar{\theta} \leftarrow \tau\theta + (1 - \tau)\bar{\theta}$.

## A.4 PSEUDOCODE: DQN TRAINING AND ONLINE EXECUTION

---

**Algorithm 2** RAP Controller Training via Masked DQN

---

**Require:** Dense model $\mathcal{M}_{\text{dense}}$; proxy corpus $\mathcal{C}$; distribution over requests $(R_{\text{bs}}, R_{\text{sql}})$ and budgets $B$; replay buffer $\mathcal{D}$; discount $\gamma$; schedule $\varepsilon_t$
1: Initialize $Q_\theta$, target $Q_{\bar{\theta}} \leftarrow Q_\theta$; initialize optimizer; set $\alpha{=}1.0, \beta{=}0.3, \eta{=}1, \zeta{=}0.1$
2: **for** episode $= 1, \ldots, E$ **do**
3:     Sample request $(R_{\text{bs}}, R_{\text{sql}})$ and budget $B$; set $\mathcal{M}_0 \leftarrow \mathcal{M}_{\text{dense}}$; $t \leftarrow 0$
4:     Run GSI to obtain initial importance scores for $s_0$; build action mask $\mathcal{A}_{\text{valid}}(s_0)$
5:     **while** $t < H$ **do**
6:         Select $a_t$ by masked $\varepsilon$-greedy from $Q_\theta(s_t, \cdot)$
7:         **if** $a_t = 0$ **then**                                              $\triangleright$ STOP
8:             Compute $r_t$ by Eq. equation 2 (with $\mathcal{M}_{t+1} = \mathcal{M}_t$), set done $\leftarrow$ True
9:         **else**
10:            $\mathcal{M}_{t+1} \leftarrow \mathcal{M}_t \setminus B_{a_t}$; recompute GSI scores; update mask
11:            Compute $r_t$ by Eq. equation 2 and done $\leftarrow \big[\text{Mem}_{\text{peak}}(\mathcal{M}_{t+1}) \leq B\big]$
12:         **end if**
13:         Store $(s_t, a_t, r_t, s_{t+1}, \text{done})$ in $\mathcal{D}$
14:         Sample a minibatch from $\mathcal{D}$; update $\theta$ by minimizing $\mathcal{L}(\theta)$; periodically update $Q_{\bar{\theta}}$
15:         **if** done **then break**
16:         **else** $t \leftarrow t + 1$
17:         **end if**
18:     **end while**
19: **end for**

---

---

**Algorithm 3** RAP Online Execution at Inference Time

---

**Require:** Trained $Q_\theta$; incoming request $(R_{\text{bs}}, R_{\text{sql}})$; measured $B = \text{Sys}_{\text{avail}}$
1: $\mathcal{M}_0 \leftarrow \mathcal{M}_{\text{dense}}$; run GSI to get initial $s_0$; $t \leftarrow 0$
2: **while** $\text{Mem}_{\text{peak}}(\mathcal{M}_t) > B$ **and** $t < H$ **do**
3:     Build $\mathcal{A}_{\text{valid}}(s_t)$; choose $a_t = \arg\max_{a \in \mathcal{A}_{\text{valid}}(s_t)} Q_\theta(s_t, a)$
4:     **if** $a_t = 0$ **then break**
5:     **end if**
6:     $\mathcal{M}_{t+1} \leftarrow \mathcal{M}_t \setminus B_{a_t}$; recompute GSI; $t \leftarrow t + 1$
7: **end while**
8: **return** pruned $\mathcal{M}_t$; run inference

---

# B DATASETS AND BASELINES

## B.1 COMMONSENE REASONING

The details of the benchmarks are as follows:

- BoolQ (Clark et al., 2019): yes/no questions which are naturally occurring and generated in unprompted and unconstrained settings. There are 3270 questions in the test set.
- PIQA (Bisk et al., 2020): questions with two solutions requiring physical commonsense. There are 1830 questions in the test set.
- HellaSwag (Zellers et al., 2019): commonsense NLI questions including a context and several endings which complete the context. There are 10042 questions in the test set.
- WinoGrande (Sakaguchi et al., 2019): fill-in-a-blank task with binary options to choose the right option for a given sentence which requires commonsense reasoning. There are 1267 questions in the test set.
- ARC-easy (Clark et al., 2018) & ARC-challenge (Clark et al., 2018): the Challenge Set and Easy Set of ARC dataset of genuine grade-school level, containing 2376/1172 multiple-choice science questions in the test set, respectively.
- OpenbookQA (Mihaylov et al., 2018): uestions requiring multi-step reasoning, use of additional commonsense knowledge, and rich text comprehension. There are 500 questions in the test set.

## B.2 Baselines

- *LLMPruner* (Ma et al., 2023a), which adopts structural pruning that selectively removes non-critical coupled structures based on weights and gradient information, maximally preserving the majority of the LLM's functionality. LLMPruner applies post training to the pruned model, but for fair comparison, we do not apply post training to it. However, LLMPuner requires extra overhead for pruning its gradient-base pruning policy.
- *SliceGPT* (Ashkboos et al., 2024), which is a post-training sparsification scheme that replaces each weight matrix with a smaller matrix, reducing the embedding dimension of the network. Specifically, they applied PCA to the hidden representation from shallow to deep layers, and incorporated the dimension reduction matrix into existing network parameters.
- *DISP-LLM* (Gao et al., 2024), which introduces a dimension-independent structural pruning scheme that breaks inter-layer width coupling. This post-training method uses gradient-based optimization via a learned hyper-network to determine which neurons to remove in each layer, enabling flexible layer-specific width reduction without additional fine-tuning.
- *ShortGPT* (Men et al., 2024) reveals significant redundancy among LLMs by proposing a layer-pruning method that removes redundant layers with minimal performance degradation
- *MHA-Drop* (He et al., 2024), which prunes entire multi-head self-attention layers of Transformer blocks to accelerate inference. By removing a fraction of the attention layers based on cosine similarity-based importance, this approach achieves notable speedups with minor impact on the model performance.
- *FFN-Skip* (Jaiswal et al., 2024), which applys inference-time skipping strategy that omits selected feed-forward network layers to reduce computation. It leverages an input-adaptive criterion to dynamically skip FFN blocks during decoding, yielding faster generation with negligible degradation in output quality.

## C More Results

Table 3 shows additional results on Qwen-1.5-7B and Qwen-2.5-7B, which confirms the proposed *RAP* is architecture-agnostic: it preserves competitive perplexity and downstream accuracy across two distinct generations of the Qwen series, implying that the same pruning strategy can be ported to other modern transformer backbones with minimal modification.

## D Limitation

Despite its promising results, *RAP* still faces several important limitations. First, the Greedy Sequential Importance procedure relies on repeated perplexity measurements over an external corpus

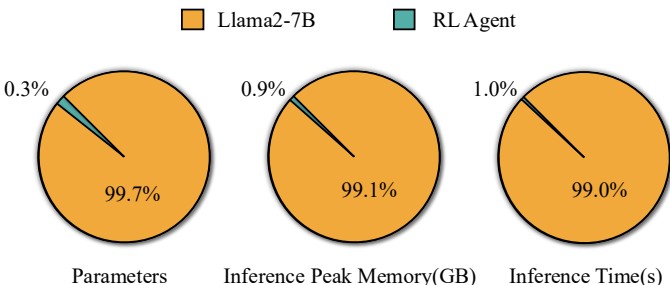

Figure 11: Overhead analysis comparing the RL agent and Llama2-7B in terms of parameter, peak memory usage, and inference latency, illustrating the negligible cost of deploying the RL controller.

Table 3: Zero-shot performance of pruned versus dense model under different memory budgets. [1] 100% memory budget indicates exceeding peak inference memory usage (parameters + KV cache).

| Budget | Schemes | Perplexity ↓ | | Commonsense Task (%) ↑ | | | | | | | |
|---|---|---|---|---|---|---|---|---|---|---|---|
| | | WikiText2 | PTB | BoolQ | PIQA | WinoG. | HellaS. | ARC-e | ARC-c | OBQA | Avg. |
| **Qwen1.5-7B** | | | | | | | | | | | |
| 100%[1] | Dense | 7.95 | 11.93 | 82.45 | 79.05 | 66.14 | 76.90 | 62.25 | 42.83 | 41.60 | 64.46 |
| 80% | ShortGPT (Men et al., 2024) | 16.88 | 24.88 | 43.98 | 72.69 | 58.41 | 59.11 | 54.50 | 33.70 | 32.20 | 50.66 |
| | MHA-Drop (He et al., 2024) | 14.26 | 22.73 | 59.91 | 75.90 | 58.96 | 67.61 | 61.73 | 41.89 | 37.20 | 57.59 |
| | FFN-Skip (Jaiswal et al., 2024) | 94.77 | 123.12 | 45.26 | 59.19 | 51.30 | 36.67 | 36.41 | 22.70 | 28.00 | 39.93 |
| | *RAP* | **18.88** | **30.88** | **64.50** | **73.39** | **56.51** | **59.98** | **56.26** | **36.09** | **38.60** | **55.05** |
| 60% | ShortGPT | 445.24 | 701.1 | 54.55 | 56.08 | 51.07 | 32.49 | 32.37 | 24.23 | 28.40 | 39.89 |
| | MHA-Drop | 628.12 | 676.62 | 45.87 | 54.45 | 51.45 | 33.16 | 33.08 | 25.67 | 29.59 | 39.05 |
| | FFN-Skip | 1889780.25 | 2455505.75 | 46.7 | 51.69 | 49.64 | 26.41 | 25.21 | 25.85 | 28.79 | 36.33 |
| | *RAP* | **54.48** | **68.33** | **54.76** | **61.70** | **51.07** | **39.72** | **44.28** | **24.32** | **29.59** | **43.64** |
| **Qwen2.5-7B** | | | | | | | | | | | |
| 100% | Dense | 6.85 | 11.36 | 84.61 | 79.71 | 73.00 | 78.95 | 77.40 | 51.01 | 47.40 | 70.30 |
| 80% | ShortGPT | 523.53 | 2154.89 | 72.20 | 66.59 | 56.35 | 48.50 | 61.99 | 40.27 | 36.40 | 54.62 |
| | MHA-Drop | 115.11 | 184.05 | 42.75 | 71.38 | 57.46 | 55.60 | 52.90 | 39.25 | 40.40 | 51.39 |
| | FFN-Skip | 141.24 | 175.33 | 48.69 | 61.26 | 53.51 | 42.08 | 45.16 | 31.14 | 30.00 | 44.55 |
| | *RAP* | **13.56** | **20.33** | **70.46** | **72.74** | **60.62** | **63.93** | **57.87** | **37.29** | **35.19** | **56.87** |
| 60% | ShortGPT | 3460.52 | 4107.47 | 38.59 | 54.03 | 52.80 | 27.61 | 26.56 | 23.63 | 25.40 | 35.52 |
| | MHA-Drop | 9099.49 | 16067.49 | 48.47 | 54.30 | 50.99 | 28.23 | 29.67 | 27.38 | 32.20 | 38.75 |
| | FFN-Skip | 1628213.25 | 1434617.50 | 45.78 | 52.12 | 48.93 | 26.83 | 24.54 | 27.3 | 27.6 | 36.16 |
| | *RAP* | **306.13** | **423.79** | **47.80** | **57.99** | **51.07** | **33.64** | **34.33** | **26.54** | **30.80** | **40.31** |

| Schemes | Llama2-7B 80% | Llama2-7B 60% | Llama3-8B 80% | Llama3-8B 60% |
|---|---|---|---|---|
| LLMPruner | 35% | 45% | 35% | 45% |
| SliceGPT | 40% | 65% | 40% | 65% |
| ShortGPT | ~37% | ~75% | ~31% | ~75% |
| MHA-Drop | ~26% | ~32% | ~12% | ~15% |
| FFN-Skip | ~52% | ~64% | ~65% | ~81% |
| *RAP* | ~24% | ~30% | ~31% | ~42% |

Table 4: The pruning ratio of model weight within the memory budget for different heuristics schemes.

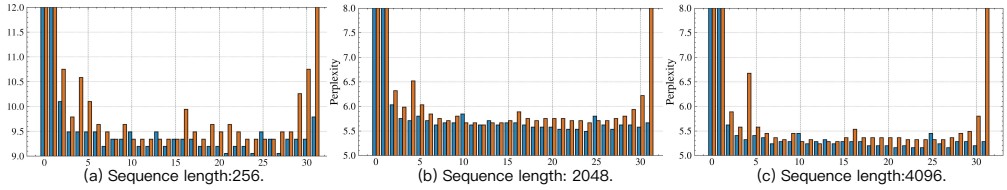

Figure 12: Block sensitivity analysis: removing specific MHA and FFN under diff. sequence length

(Alpaca), which may become computationally prohibitive for models with tens-of-billions of parameters or for domains lacking a representative calibration set, thereby limiting scalability. Secondly, while the online controller adds negligible inference latency, the offline reinforcement-learning stage still demands several hundred seconds of GPU time and shows sensitivity to the reward coefficients $\alpha, \beta$, suggesting non-trivial tuning effort for new hardware or workload profiles. Thirdly, the current state representation tracks only batch size, sequence length and instantaneous memory, omitting

latency, energy and heterogeneous device characteristics; as a result, the learned policy may yield sub-optimal trade-offs when such factors dominate deployment objectives. Finally, we note that addressing the challenges of long-context inference, which leads to substantial growth in the KV cache and is often infeasible on resource-constrained devices, is beyond the scope of this paper. Nevertheless, we believe our method's demonstrated efficiency in compressing the KV cache provides a promising foundation for future community efforts in long-context inference compression.

## E   THE USE OF LARGE LANGUAGE MODELS

We used LLMs solely as a writing-assistance tool to polish our paper (grammar, wording, concision, and minor LaTeX formatting). The LLM did not contribute to research ideation, problem formulation, method design, experiments, data analysis, results, or conclusions, and it was not used to generate citations or technical content. All suggestions were reviewed and, when adopted, edited by the authors, who take full responsibility for the paper's content; no proprietary data beyond the manuscript text was shared with the tool.

