# OpenReview forum: "Runtime Adaptive Pruning for LLM Inference"
_ICLR.cc/2026/Conference — Submitted to ICLR 2026_

### Official Review · Reviewer_Hb4M · 2025-10-28

**Soundness:** 2
**Presentation:** 2
**Contribution:** 2
**Rating:** 2
**Confidence:** 3

**Summary:**

This paper proposes RAP (Runtime-Adaptive Pruning), a reinforcement learning-based framework for dynamically pruning Large Language Models (LLMs) during inference. Unlike existing static pruning methods, RAP adapts pruning decisions based on runtime conditions including memory constraints, batch sizes, and sequence lengths. The framework uses Greedy Sequential Importance (GSI) analysis to iteratively evaluate block importance and an RL agent to select which transformer blocks to prune. While the paper shows improvements over baselines on Llama and Qwen models, there are significant concerns about the evaluation methodology and practical applicability.

**Strengths:**

1. The paper effectively demonstrates that memory bottlenecks shift dynamically between parameters and KV cache depending on workload characteristics.This is a valuable insight for the community.
2. The evaluation covers multiple model families (Llama2, Llama3, Qwen) and includes both generation quality metrics and downstream task performance across 7 benchmarks.
3. The three key observations in Section3 provide compelling evidence for adaptive approaches, particularly the insight that KV cache becomes the dominant memory bottleneck at larger batch sizes and sequence lengths.

**Weaknesses:**

1. The paper fails to properly isolate the RL agent's contribution. While it compares against one-shot GSI and random selection, it critically lacks comparison with iterative GSI without RL (i.e., greedily removing blocks with highest GSI scores and re-evaluating). The comparison to random selection is hardly competitive and doesn't demonstrate the value of the learned policy. Without this ablation, it's unclear whether the RL agent adds any value over simply following GSI scores greedily.

2. Computing GSI requires running inference through the full model initially, meaning the method can only be deployed on machines with enough memory for the unpruned model. This defeats the primary purpose - you cannot use RAP on memory-constrained devices where the full model doesn't fit, which is exactly where such methods are most needed. This is a critical flaw that severely limits practical deployment.

3. The paper claims minimal overhead by focusing on the RL agent's 18K parameters, but completely ignores that GSI requires multiple forward passes through the model for importance evaluation. Each GSI iteration requires a full forward pass, making the actual latency overhead potentially orders of magnitude higher than reported. This is a serious misrepresentation of the computational cost.

4. The paper doesn't clarify which baseline methods were designed to handle parameters + KV cache pruning vs just parameter pruning. If baselines only prune parameters while RAP prunes both, this creates an unfair advantage. Even if all methods prune parameters + KV cache but baselines were not designed to work for KV cache, it is still an unfair comparison. The evaluation protocol makes KV cache pruning the most important component, potentially biasing results if baselines aren't designed for this

5. The paper's valuable contribution - demonstrating that "KV cache pruning is an important part of model pruning for memory optimization" - is buried in the experimental analysis rather than being a central claim. This insight about the dominance of KV cache in memory bottlenecks could be the paper's great contribution if properly emphasized.

6. GSI is essentially iterative pruning with re-evaluation (a known technique), and the RL formulation uses standard DQN without innovations. The combination doesn't justify the complexity when simpler approaches might work equally well.

7. No evaluation on truly dynamic workloads despite "runtime-adaptive" claims

**Questions:**

1. Can you provide results for iterative GSI without RL (greedily selecting highest GSI scores with re-evaluation)? This is essential to understand if the RL agent adds value beyond following GSI scores.

2. How do you address the fundamental issue that GSI requires full model memory initially? Can the method work on devices where the full model doesn't fit? If the target deployment device cannot run the full model, how can RAP be used at all given GSI's requirements?

3. What is the actual wall-clock latency overhead including all GSI computations? How many forward passes are required in practice?

4. Please provide a clear table showing: 1/ which baseline methods prune parameters only vs parameters + KV cache, 2/ which methods were originally designed for KV cache pruning, 3/ whether you modified any baselines to handle KV cache

---

> ### Author Response · Authors · 2025-11-27
> **Response to Reviewer Hb4M**
>
> We sincerely thank for your detailed and insightful feedback. We value the opportunity to clarify the distinction between our **GSI** and **RL**, and to further demonstrate the benefits of our method.
>
> Below, we address each weakness and question point-by-step.
>
> ## ***W1 & Q1. Isolation of RL contribution and lack comparison of GSI without RL***
>
> We appreciate the reviewer pointing this out. There appears to be a misunderstanding regarding the role of **GSI** vs **RL**. Iterative GSI alone cannot address ***runtime adaptation*** issue. Only GSI would always remove blocks in the same fixed order **regardless of** workload conditions. The RL agent's value lies in:
>
> > 1. Memory-aware decision-making: The state of RL includes $s_t^{Sys} = (Sys_{avail}, Sys_{req})$ (Section 4.2), enabling it to stop pruning when the budget is satisfied.
>
> > 2. Request-conditioned pruning: The state also encodes $s_t^{Req} = (R_{bs}, R_{sql})$. As Figure 4 demonstrates, block importance varies significantly across sequence lengths and even the same block ranking is suboptimal across heterogeneous requests.
>
> > 3. Balancing dual objectives: The reward function (Eq. 2) explicitly trades off performance preservation against memory efficiency, which greedy GSI cannot jointly optimize.
>
> ## ***W2 & Q2.  Full-model memory requirement for GSI***
>
> We apologize for the confusion and want to clarify **GSI is an offline calibration step**. It is run **once per model** on a calibration corpus (Alpaca), on a machine where the dense model fits, to compute per-block importance. The resulting importance scores $MHA_{imp,i}$, $FFN_{imp,i}$ are then stored and used as part of the RL state $s_t^{\text{Model}}$ at inference time.
>
> Moreover, unlike prior methods[1-3] introduce additional memory overhead at each pruning operation, our approach only requires a one-time offline calibration cost. The subsequent pruning stage operates with no additional memory cost, making our approach highly practical for deployment on resource-constrained edge devices.
>
> [1] LLM-Pruner: On the Structural Pruning of Large Language Models. NeurIPS 2023
> [2] SliceGPT: Compress Large Language Models by Deleting Rows and Columns. ICLR 2024
> [3] DISP-LLM: Dimension-Independent Structural Pruning for Large Language Models. NeurIPS 2024
>
> ## ***W3 & Q3. GSI Computational Overhead***
>
> GSI is used solely during the offline training phase of the RL agent to generate reward signals and is reusable for all future requests in online inference. The specific training time depends on practical requirements. For simple deployment scenarios, rapid calibration on small-scale data is sufficient. In our experiments, to make the compressed model more competitive, we selected the larger Alpaca dataset for fine-grained calibration. On the LLaMA-3.1-8B model, calibration takes approximately 4 hours using a single NVIDIA A40 GPU. Considering the ';**"once-for-all"** characteristic of our method, this additional cost is entirely acceptable.

---

> > ### Author Response · Authors · 2025-11-27
> > **Follow-up Response to Reviewer Hb4M**
> >
> > ## ***W4 & Q4. Fair Comparison of KV cache pruning across baselines***
> >
> > We provide the requested clarification:
> >
> > | Method    | Prunes parameters            | Prunes KV cache              | Originally designed for KV pruning? | Runtime-adaptive |
> > | --------- | ---------------------------- | ---------------------------- | ----------------------------------- | ---------------- |
> > | LLMPruner | Yes                          | Yes (via MHA heads pruning)  | No                                  | No               |
> > | SliceGPT  | Yes                          | Yes (via MHA heads pruning)  | No                                  | No               |
> > | DISP-LLM  | Yes                          | Yes (via MHA heads pruning)  | No                                  | No               |
> > | ShortGPT  | Yes                          | Yes (via layer pruning)      | Yes (parameters + KV)               | No               |
> > | MHA-Drop  | Yes                          | Yes (via MHA blocks pruning) | Yes (MHA-oriented)                  | No               |
> > | FFN-Skip  | Yes (via FFN blocks pruning) | No                           | No                                  | No               |
> > | RAP       | Yes (via FFN blocks pruning) | Yes (via MHA blocks pruning) | Yes (parameters + KV)               | Yes              |
> >
> > We would like to emphasize that the motivation of RAP is to address the imbalanced resource consumption of parameters and KV cache across different requests, as illustrated in Figure 3. The "unfair advantage" concern is actually the core motivation of our work: existing methods were not designed to jointly optimize both memory components, leading to suboptimal performance under realistic memory budgets. One of our key contributions is jointly optimization via reducing memory consumption from parameters by removing entire FFN blocks, while simultaneously reducing both parameter memory and KV cache overhead from the attention mechanism by removing entire MHA blocks. Among the baselines, LLMPruner[1], SliceGPT[2] and DISP-LLM[3] focus on weight reduction. When they remove a head from the MHA layer, they implicitly reduce KV cache, but do not optimize for it dynamically. ShortGPT[4], MHA-Drop[5], and FFN-Skip[6] compress models by removing entire MHA and FFN blocks, but lack adaptivity to varying inputs and resource constraints.
> >
> > [1] LLM-Pruner: On the Structural Pruning of Large Language Models. NeurIPS 2023
> > [2] SliceGPT: Compress Large Language Models by Deleting Rows and Columns. ICLR 2024
> > [3] DISP-LLM: Dimension-Independent Structural Pruning for Large Language Models. NeurIPS 2024
> > [4] ShortGPT: Layers in Large Language Models are More Redundant Than You Expect. arXiv 2024
> > [5] What Matters in Transformers? Not All Attention is Needed. arXiv 2024
> > [6] FFN-SkipLLM: A Hidden Gem for Autoregressive Decoding with Adaptive Feed Forward Skipping. EMNLP main 2024
> >
> > ## ***W5. Emphasis on KV Cache Insight***
> >
> > As shown in Figure 3, KV cache becomes the dominant source of memory consumption when batch size and request sequence length are large. Conversely, when these factors are small, model parameters dominate memory overhead. This observation motivates the core contribution of our work: jointly optimizing parameters and KV cache in a request-aware and resource-adaptive manner, rather than focusing solely on KV cache compression.
> >
> > ## ***W6. Limited Technical Novelty***
> >
> > We respectfully disagree that the combination lacks novelty:
> >
> > > 1. GSI addresses a specific limitation of one-shot pruning: inter-layer dependencies cause cascading errors (Figure 6 shows one-shot methods miss heterogeneity)
> > > 2. The RL formulation is non-trivial: The action space of $2^{64}$ is decomposed into sequential decisions with memory-aware masking which is a meaningful design choice, not vanilla DQN application
> > > 3. The problem formulation is novel: No prior work frames LLM pruning as runtime-adaptive compression with joint parameter+KV optimization
> >
> > That said, we acknowledge the individual components are not groundbreaking but the contribution lies in the system-level integration enabling practical runtime-adaptive deployment.

---

> > > ### Author Response · Authors · 2025-11-27
> > > **Follow-up Response to Reviewer Hb4M**
> > >
> > > ## ***W7. Evaluation on dynamic workloads***
> > >
> > > Our motivation and RL formulation are explicitly built around dynamic workloads: §3 uses an Azure LLM-service trace to show that request lengths and arrival rates fluctuate substantially over time (Figure 2), and §2.1 shows how this induces sharp transitions between parameter- and KV-dominated regimes (Figure 3).
> > >
> > > During RL training, each episode samples varying batch sizes, sequence lengths, and memory budgets from this workload distribution, and the reward penalizes budget violations and perplexity degradation. The learned policy thus conditions its actions on per-request runtime states rather than a fixed budget. Our experiments at 60% and 80% memory budgets serve as representative samples from a spectrum of dynamic workloads, illustrating how our method adapts to varying resource constraints in practice.

---

### Official Review · Reviewer_mLUh · 2025-10-29

**Soundness:** 3
**Presentation:** 2
**Contribution:** 2
**Rating:** 4
**Confidence:** 3

**Summary:**

The paper proposes RAP, a runtime-adaptive pruning framework that (i) computes Greedy Sequential Importance scores for FFN/MHA blocks; and (ii) uses a lightweight RL controller to pick a pruning policy per request under a memory budget that includes both parameters and KV-cache.

**Strengths:**

1. Considering parameters and KV cache as the target is novel as most pruning work optimizes only weights.

2.  Design with MLP with small overhead makes it easy and efficient to deploy.

3. The ablation study shows the effectiveness of this method.

**Weaknesses:**

1. Only zero-shot short-answer benchmarks. But long-context tasks (where KV matters) or real generation quality would better show the purported advantage.

2. Need end-to-end latency and throughput comparison.  Real-world servers aslo care about tokens/sec and tail latency besides memory savings.

3. If heads or layers are dropped at runtime, how are pre-existing KV tensors handled across decoding steps?

4. If GSI already orders blocks and the agent “iteratively removes the least important,” where does RL refine this order?

**Questions:**

See Weaknesses.

---

> ### Author Response · Authors · 2025-12-04
> **Response to Reviewer mLUh**
>
> We sincerely thank for your constructive feedback. We appreciate the emphasis on practical deployment metrics and the request for clarification on the interaction between GSI ranking and the RL policy. These insights have helped us strengthen the validation of our real-world applicability.
>
> Below, we address the weaknesses and questions point-by-point.
>
> ## ***W1 & Q1. Evaluation on Long-Context and Generative Tasks***
>
> We appreciate the feedback from the reviewer regarding the evaluation on long context tasks. We respectfully submit that our method RAP is specifically designed for resource constrained edge devices and real time mobile applications. These hardware platforms typically operate under strict thermal and battery limitations which naturally restrict the feasibility of processing extremely long text sequences regardless of the memory optimization strategy. Therefore our design primarily targets sequence lengths that are practical and common for deployment on such edge devices.
>
> Moreover our current evaluation protocol does include validation on tasks that require handling longer dependencies and generation quality. We employ standard generative benchmarks such as WikiText2 and PTB to assess the capability of the model beyond simple short answer questions. These datasets necessitate maintaining coherence over continuous text streams and serve as effective proxies for real generation quality in our target scenarios. In addition we have conducted sensitivity analyses across varying sequence lengths to demonstrate that RAP maintains robustness as input length increases within the operational range of edge hardware.
>
> ## ***W2 & Q2. End-to-End Latency and Throughput Comparison***
>
> We appreciate the feedback regarding system level metrics. We wish to clarify that our proposed method is primarily tailored for resource constrained edge devices rather than high performance server environments. On such edge platforms the strict memory limit often serves as the hard constraint that dictates feasibility whereas servers prioritize aggregate throughput. To address the concern regarding practical efficiency we have conducted additional experiments to evaluate the end to end latency and throughput on a representative edge device. The results presented below demonstrate that our approach achieves lower latency and higher throughput compared to baselines by effectively reducing memory consumption and computational overhead.
>
> | Method | Latency (s) | Throughput (token/s) |
> |:---------|---------:| ---------:|
> | Dense  | 52.40  | 39.08 |
> | LLM-Pruner  | 83.47   | 24.53 |
> | SliceGPT   | 64.64   | 31.68 |
> | ***RAP***   | 43.36   | 47.23 |
>
> ## ***W3 & Q3. Handling KV Tensors for Runtime Drop***
>
> This is an important implementation detail. We clarify it here and will add a paragraph Appendix in the revised version.
>
> For each request, RAP first observes the request statistics and current available memory, then runs the RL controller to select a pruning policy (which FFN/MHA blocks to keep or drop) before decoding any tokens for that request. The pruned block will be offload to CPU memory and can be restored for new quest.
>
> The chosen sub-architecture is then used consistently for all decoding steps of that request. KV caches are allocated only for the surviving attention blocks. Dropped blocks are never instantiated, so there are no KV entries for them.
>
> For following new request, the RL agent generates a new pruning strategy based on the request characteristics and resource availability, deciding whether to offload blocks to CPU or load blocks from CPU to GPU, while clearing all KV cache from the previous request. This request-level adaptation mechanism enables RAP to dynamically accommodate varying runtime deployment environments.
>
> ## ***W4 & Q4. Role of RL vs. GSI***
>
> We appreciate this question and give a sufficient and clear explanation here.
>
> GSI provides a static ranking of block importance; however, the RL reward jointly considers both block importance and available memory. Consequently, when making decisions, the RL agent balances performance (block importance) and resources (available memory) to produce a dynamic and holistic block ranking, rather than relying solely on the static importance ranking from GSI. Specifically, under tight memory constraints, the RL agent tends to retain blocks with lower importance but smaller resource footprints, while removing blocks with higher importance but larger overhead. This design powers RAP adaptive pruning capabilities across varying memory budgets and real-time requests.

---

### Official Review · Reviewer_SaEz · 2025-11-01

**Soundness:** 2
**Presentation:** 3
**Contribution:** 2
**Rating:** 6
**Confidence:** 4

**Summary:**

This paper proposes RAP,  framework for the runtime-adaptive pruning of Large Language Models (LLMs) based on RL. The core motivation is that real-world inference workloads and system memory availability are highly dynamic, which static pruning strategies cannot accommodate. RAP introduces a Greedy Sequential Importance (GSI) algorithm to better assess block importance and an RL agent that observes real-time system state and request characteristics to dynamically select which MHA or FFN blocks to prune, covering input-driven and system-level variance. Experiments show that RAP outperforms static pruning baselines under this budget-aware evaluation.

**Strengths:**

- The shift from evaluating at a "fixed sparsity ratio" to a "fixed memory budget" is interesting. It more accurately reflects the deployment constraints on resource-limited devices.
- The results clearly show that RAP makes more intelligent pruning decisions than static baselines, especially under aggressive memory budget.
- The paper includes thorough ablation studies that demonstrate the necessity of both the GSI component and the RL agent.

**Weaknesses:**

-  The entire motivation is built on optimizing for a Memory Budget. However, it fails to compare against the most effective and widely-adopted technique post-training quantization (e.g., INT4). A simple INT4 quantized model would occupy a smaller memory footprint than RAP's pruned FP16 model under the same budget.

- In real-world applications, FP16 would not be deployed.  A convincing demonstration of RAP's value would be to show that it can further reduce memory on top of a quantized model.

- The core idea is training a single and adaptive policy to replace a collection of static configurations. This concept was explored by the Once-for-All. The authors should discuss the connection to Once-for-All.

- While the memory budget focus is practical, the complete absence of a standard fixed-sparsity comparison makes it difficult to isolate and appreciate the core algorithmic improvement.

Once-for-All: Train One Network and Specialize it for Efficient Deployment. ICLR 2020

**Questions:**

- How does RAP compare to a strong INT4 quantization baseline?  Can you show the performance of "INT4 + RAP" to demonstrate complementary benefits?

- What is the computational cost of GSI?

- I am curious whether, at the same parameter sparsity level (e.g., 30%), does RAP's GSI-based policy yield higher accuracy than a one-shot importance scoring method?

---

> ### Author Response · Authors · 2025-12-04
> **Response to Reviewer SaEz**
>
> Thank you for your constructive feedback. We appreciate the suggestion to compare against and combine our method with quantization, as well as the pointer to the "Once-for-All" paradigm. These comments have helped us strengthen the practical positioning of RAP.
>
> Below, we address the weaknesses and questions point-by-point.
>
> ## ***W1 & W2 & Q1. Missing comparison with post-training INT4 quantization and INT + RAP***
>
> We appreciate the reviewer highlighting the relation between our method and post training quantization.
>
> Our work focuses on **structured pruning** under a unified memory budget, treating pruning as one of the three main compression families together with knowledge distillation and quantization, as stated in the introduction and related work. RAP removes entire attention and feed forward blocks in a runtime adaptive way which directly changes the network structure and the amount of KV cache that must be stored for each request. In contrast, post training quantization such as INT4 changes the numeric precision of parameters and activations while keeping the network structure fixed. Because these two techniques operate along different axes structure versus precision a direct head to head comparison is not the primary focus of this paper and is not needed to validate the contribution of a runtime adaptive pruning policy under fixed precision.
>
> For this reason all experiments of intentionally keep the numeric format constant FP16 for both the dense models and **all pruning baselines** so that any change in accuracy and memory usage can be attributed solely to which blocks are removed and how the pruning policy adapts to runtime memory and workload variation, rather than to differences in quantization quality or calibration procedures. This mirrors the standard evaluation setting in prior pruning work where the goal is to compare structural compression strategies at a common precision level.
>
> At the same time RAP is fully complementary to quantization. The framework only requires access to model parameters and a memory model and does not assume any particular numeric format. In practice one can either
>
> > 1. first apply an INT4 post training quantization method to obtain a quantized backbone and then run GSI and train the RAP controller on this quantized model, or
>
> > 2. first use RAP to find a pruned architecture that respects a target unified memory budget and then apply INT4 quantization to the resulting smaller model.
>
> In both cases quantization and pruning would combine multiplicatively in terms of memory saving since RAP reduces the number of active blocks and hence the size of the KV cache while INT4 reduces the bytes per stored scalar. We therefore expect that an INT4 plus RAP configuration would achieve a strictly smaller footprint than either technique alone under the same quality target.
>
> In the revision we will add a paragraph in the methodology section and the discussion that explicitly frames RAP as a pruning method that is orthogonal and complementary to post training quantization, and we will note the combined INT4 plus RAP setting as an important direction for follow up experiments and future work.
>
> ## ***W3. Connection to "Once-for-All" paper***
>
> Thanks for your pointing out and we highly appreciate this insight. "Once-for-All"[1] and RAP share the high-level goal of training a single model that adapts to different resource constraints. We will add a discussion in §3 to clarify their relationship:
>
> * Similarity: Like "Once-for-All", RAP trains a tiny model that allows sub-network selection without retraining for every scenario.
> * Distinction: "Once-for-All" typically focuses on static deployment, selecting a specific architecture in weight-sharing way once for a specific hardware device. RAP focuses on dynamic runtime inference, selecting a specific subset of LLM by block pruning for per request based on current runtime conditions. This runtime adaptivity is crucial for LLM serving where workload characteristics vary significantly across requests. In Summary, RAP is **"Once-for-All-Requests-and-Workload"** rather than "Once-for-All-Devices."
>
> [1] Once-for-All: Train One Network and Specialize it for Efficient Deployment. NeurIPS 2020

---

> > ### Author Response · Authors · 2025-12-04
> > **Follow-up Response to Reviewer SaEz**
> >
> > ## ***W4. Fixed-Sparsity Comparison***
> >
> > We focused on **memory budget** because prior methods only consider sparsity ratio, and we observed inconsistencies between sparsity ratio and actual inference memory during our implementation of baseline methods (e.g., 30% sparsity does not equate to 30% memory reduction). Therefore, RAP targets actual memory budget rather than superficial sparsity ratio, which better aligns with practical deployment requirements. However, we agree that a fixed-sparsity comparison is necessary to isolate the algorithmic effectiveness. We have provided this comparison at **30%** sparsity ratio on **Llama2-7B** in the table below.
> >
> > |  Method  |     HellaS.     |      PIQA      |      CoQA      |      BoolQ      |     Race-H     |     Race-M     |      MMLU      |      CMMLU      |      Avg.      |
> > | :-------: | :-------------: | :-------------: | :-------------: | :-------------: | :-------------: | :-------------: | :-------------: | :-------------: | :-------------: |
> > |   Dense   |      71.26      |      77.91      |      64.62      |      71.62      |      35.71      |      34.19      |      45.39      |      32.92      |      47.78      |
> > | LLMPruner | **56.46** | **71.22** |      42.51      |      55.20      |      22.56      |      22.35      |      23.33      |      25.25      |      34.78      |
> > | SliceGPT |      50.27      |      66.21      |      41.36      |      38.32      |      21.07      |      21.66      |      28.92      |      25.37      |      32.84      |
> > |    RAP    |      53.02      |      66.43      | **47.99** | **74.71** | **32.25** | **35.17** | **43.96** | **32.25** | **41.24** |
> >
> > ## ***Q2. Computational cost of GSI***
> >
> > For Llama2-7B with 64 blocks, GSI requires evaluating perplexity after each block removal. In our implementation, GSI computes block importance once per model on a large calibration corpus (Alpaca), taking approximately one-time cost of 4 hours on a single Nvidia A40 GPU. These scores are then stored and **reused** for all subsequent inferences. Your can also choose simple corpus for saving less computational overhead. We provide more information in following table.
> >
> > | Model      | Total Blocks | GSI Time (h) |
> > | :--------- | :----------: | :----------: |
> > | Llama2-7B  |      64      |      4      |
> > | Llama3-8B  |      64      |     3.9     |
> > | Qwen1.5-7B |      64      |     4.5     |
> > | Qwen2.5-7B |      56      |     3.5     |
> >
> > ## ***Q3. GSI vs. One-shot at Same Sparsity Level***
> >
> > To demonstrate the algorithmic superiority of our GSI-based policy over standard One-Shot pruning, we present the performance at fixed sparsity levels.
> >
> > | Sparsity Ratio | Method | BoolQ | PIQA | WinoG. | HellaS. | ARC-e | ARC-c | OBQA |
> > |:---:|:---:|:---:|:---:|:---:|:---:|:---:|:---:|:---:|
> > | **20%** | One-shot | 39.72 | 65.61 | 53.20 | 43.86 | 48.91 | 27.30 | 28.79 |
> > | | **GSI** | **62.81** | **73.99** | **63.38** | **65.77** | **60.35** | **36.69** | **36.60** |
> > | **30%** | One-shot | 38.37 | 52.23 | 48.86 | 27.88 | 29.38 | 24.83 | 25.8 |
> > | | **GSI** | **57.16** | **58.26** | **53.75** | **37.81** | **37.79** | **26.02** | **30.20** |

---

### Official Review · Reviewer_oDjU · 2025-11-27

**Soundness:** 3
**Presentation:** 2
**Contribution:** 2
**Rating:** 2
**Confidence:** 4

**Summary:**

This paper identifies a key limitation of existing LLM pruning approaches, namely their reliance on static importance criteria that fail to reflect the dynamic workload variations encountered in real-world deployment. To address this issue, the authors propose RAP (Runtime-Adaptive Pruning), a reinforcement learning-based framework that adjusts the degree of structured pruning according to runtime conditions.

The authors argue that conventional one-shot pruning evaluates block importance independently under a fixed model structure, thereby neglecting strong inter-layer dependencies within Transformers and leading to cumulative performance degradation. To mitigate this, they introduce Greedy Sequential Importance (GSI), which sequentially removes blocks while re-evaluating perplexity at each step.

RAP further incorporates runtime signals such as batch size, sequence length, system memory, and KV-cache ratio as state inputs to an RL policy that determines block removal decisions. As a result, pruning intensity varies across requests, allowing the model to adapt its structure to meet memory constraints while minimizing degradation in perplexity and downstream task performance. Through block-level structured pruning, the framework removes entire parameter and attention components, indirectly reducing KV cache usage.

Experimental results show that RAP achieves significantly lower perplexity and better commonsense reasoning performance compared to random drop and static pruning baselines under identical memory budgets, with the combination of GSI and RL yielding the most stable performance–compression trade-off.

**Strengths:**

* Clearly formulates the problem of runtime-aware pruning in realistic LLM inference settings (varying input length, batch size, and memory constraints) and convincingly highlights the limitations of static pruning approaches.
* Introduces Greedy Sequential Importance (GSI) as a principled mechanism that accounts for inter-layer dependencies through sequential importance re-evaluation, effectively mitigating performance degradation associated with one-shot pruning.
* Employs an RL-based controller to dynamically adjust pruning strength on a per-request basis, enabling adaptive responses to heterogeneous workload conditions.
* Adopts structured block-level pruning, enhancing hardware efficiency and practical deployability.
* Demonstrates empirical performance across multiple models and memory budgets, showing superior perplexity retention compared to heuristic baselines.

**Weaknesses:**

* The construction of the calibration set used for GSI computation in Table 1 is insufficiently specified, making it difficult to rule out the possibility that benchmark test data distributions were indirectly utilized during pruning, raising concerns regarding the fairness of the evaluation protocol.
* Since GSI is repeatedly recomputed based on a proxy corpus and sampled request distributions, the stability and reproducibility of the resulting importance scores remain unclear, potentially introducing variability in pruning outcomes.
* Although GSI is framed as an offline calibration step using a proxy corpus, generating an optimal pruning model for specific user scenarios would in practice require full-model access and repeated GSI execution, for which the associated computational overhead is not sufficiently quantified or analyzed.
* Despite claims regarding edge deployment feasibility, the necessity of full-model access and GSI computation during the pruning phase suggests a non-trivial operational gap between the proposed framework and practical deployment constraints.
* The pruning mechanism is inherently irreversible within an inference session, preventing mid-generation structural readjustments, which limits adaptiveness in multi-turn or reasoning-heavy scenarios where block importance may shift over time.
* The framework does not introduce an explicit KV-cache-specific pruning strategy, instead relying on indirect reduction through block removal, thereby weakening the claim of joint optimization between model parameters and KV cache memory.

**Questions:**

See Weaknesses.

---

> ### Author Response · Authors · 2025-12-04
> **Response to Reviewer oDjU**
>
> Thank you for your thorough and constructive review. We appreciate the detailed feedback and address each concern below.
>
> ## ***W1 & Q1. Calibration Set Construction and Fairness***
>
> As briefly mentioned in Section 5.1, we use the Alpaca dataset[1] for the GSI importance computation and RL reward calculation. The evaluation benchmarks (WikiText2, PTB, BoolQ, etc.) are never seen by the model during the GSI computation or RL policy training phases. Moreover, the Alpaca dataset consists of instruction-following data, which is significantly distinct from the language modeling tasks and commonsense reasoning tasks used for testing.
>
> Beyond avoiding the concerns raised above, our design offers the additional flexibility of choosing calibration datasets tailored to specific applications. This allows the block importance scores computed by GSI to be more sensitive to downstream tasks.
>
> [1] Stanford Alpaca: An Instruction-following LLaMA model.
>
> ## ***W2 & Q2. Stability and Reproducibility of GSI***
>
> We acknowledge this concern and provide the following clarifications:
>
> Given a fixed calibration corpus and model, GSI produces deterministic importance scores. At each step, it select the block whose removal yields minimum perplexity increase, enabling it fully deterministic. Furthermore, figure 9 in our paper demonstrates that RL training converges consistently across three random seeds, indicating stable GSI-derived importance scores.
>
> ## ***W3 & Q3. Computational Overhead of GSI***
>
> We do not consider "once training for one scenario" to be a weakness. On the contrary, compared to baseline methods that require calibration for each pruning operation, our one-time cost is more practical and economical for real-world deployment. The table below shows the time required for computing GSI:
>
> | Model      | Total Blocks | GSI Time (h) |
> | :--------- | :----------: | :----------: |
> | Llama2-7B  |      64      |      4      |
> | Llama3-8B  |      64      |     3.9     |
> | Qwen1.5-7B |      64      |     4.5     |
> | Qwen2.5-7B |      56      |     3.5     |
>
> ## ***W4 & Q4. Operational Gap Between Framework and Edge Deployment***
>
> We clarify that our framework is designed to explicitly decouple the resource-intensive calibration from the lightweight deployment. The GSI mechanism is strictly a one-time offline calibration step performed on a resource-sufficient to derive block importance scores ($MHA_{imp,i}$, $FFN_{imp,i}$). On the edge device, these scores are merely loaded as part of the static RL state $s_t^{\text{Model}}$, requiring zero additional memory or computation. Unlike prior methods [1-3] which introduce runtime memory overheads for pruning operations, our separation of offline calibration and online execution ensures that the framework bridges the gap to resource-constrained edge devices without operational friction.
>
> [1] LLM-Pruner: On the Structural Pruning of Large Language Models. NeurIPS 2023
> [2] SliceGPT: Compress Large Language Models by Deleting Rows and Columns. ICLR 2024
> [3] DISP-LLM: Dimension-Independent Structural Pruning for Large Language Models. NeurIPS 2024
>
> ## ***W5 & Q5. Irreversibility within Inference Sessions***
>
> We appreciate the reviewer's comment. However, we respectfully argue that this is not a weakness but rather a deliberate design choice. First, the pruning mechanism is intentionally irreversible within a single session: for each individual user request, the RL agent derives pruning decisions based on the request characteristics and resource availability, and structural readjustment during generation is unnecessary. Second, for each subsequent request, RAP generates a new pruning decision. Moreover, we provide a block restore mechanism that can reload previously offloaded blocks from CPU back to GPU when needed. This design ensures that RAP remains adaptive to multi-turn dialogues and reasoning-intensive scenarios.
>
> ## ***W6 & Q6. No Explicit KV Cache-Specific Pruning Strategy***
>
> We appreciate this insightful critique and provide clarification:
>
> Our framework achieves KV cache reduction through MHA block removal, which eliminates the corresponding KV cache entries. Our claim of 'joint optimization' refers to the RL agent's ability to consider both parameter memory (dominated by FFN) and KV cache memory (dominated by MHA) when making pruning decisions, adapting the MHA/FFN pruning ratio based on runtime workload characteristics.

---

### Meta-Review · Area_Chair_TKzG · 2026-01-05

**Summary:**

This submission introduces RAP (Runtime-Adaptive Pruning), a reinforcement learning-based framework designed to address dynamic memory budgets and varying workload demands during LLM inference. The primary contribution is the joint optimization of model weights and KV-cache through a Greedy Sequential Importance (GSI) calibration and an RL-driven controller, which offers a practical perspective on deploying LLMs on resource-constrained edge devices.

Regarding the review process, the authors raised significant concerns about the integrity of certain feedback. The AC observed that several reviews  lacked technical depth and appeared to be AI-generated.

However, even focusing on the substantive human-expert feedback, several critical concerns remain unresolved, including insufficient long-context validation and unclear detail of handling KV cache at runtime. While the idea of runtime adaptivity is promising, the current evidence does not sufficiently address these fundamental limitations. Therefore, the recommendation is Reject.

**Reviewer Concerns:**

#### **Addressed by Rebuttal**

* **Standard Comparisons:** The authors added fixed-sparsity benchmarks (e.g., Llama2-7B at 30% sparsity), showing RAP's GSI policy outperforms one-shot pruning and SliceGPT.
* **Calibration Overhead:** Clarified that GSI is a one-time offline cost (~4 hours), decoupling training time from inference latency.
* **Policy Clarification:** Resolved confusion regarding the RL agent's deterministic per-request pruning mechanism and its interaction with the GSI ranking.

#### **Outstanding Concerns**

* **Quantization Baseline:** The authors argue pruning is orthogonal to quantization, but they did not provide a direct empirical comparison. For edge deployment, **INT4 quantization** remains the primary baseline; without a Pareto front comparison, RAP’s practical advantage is unproven.
* **Long-Context Validation:** Despite claiming KV-cache optimization, evaluation remains limited to short-sequence benchmarks. There is no data proving RAP maintains generation quality in long-context tasks where KV-cache pressure is actually a bottleneck.
* **Hardware Realism:** The "block restore" mechanism (CPU-GPU swapping) lacks analysis regarding PCIe bandwidth bottlenecks, leaving the claim of practical "edge deployability" insufficiently supported.

**Reviewer Scores:**

Likely not.

---

### Decision · Program_Chairs · 2026-01-26

Reject